# Resident-Related Factors Influencing Antibiotic Treatment Decisions for Urinary Tract Infections in Dutch Nursing Homes

**DOI:** 10.3390/antibiotics11020140

**Published:** 2022-01-21

**Authors:** Lisa Marie Kolodziej, Sacha Daniëlle Kuil, Menno Douwe de Jong, Caroline Schneeberger

**Affiliations:** Department of Medical Microbiology, Amsterdam UMC, University of Amsterdam, 1105 AZ Amsterdam, The Netherlands; s.d.kuil@amsterdamumc.nl (S.D.K.); m.d.dejong@amsterdamumc.nl (M.D.d.J.); carolineschneeberger@gmail.com (C.S.)

**Keywords:** urinary tract infection, nursing homes, antibiotic prescribing

## Abstract

The aim of this cohort study was to identify resident-related factors that influence antibiotic treatment decisions for urinary tract infections (UTIs) in nursing home residents and to provide an overview of the appropriateness of antibiotic treatment decisions according to the updated Dutch guideline for UTIs in frail older adults. The PROGRESS study dataset, consisting of 298 suspected UTI episodes in Dutch nursing home residents, was used. The presence of dysuria was associated with the highest frequency of antibiotic prescription (87.8%). Positive leukocyte esterase dipstick results showed the greatest increase in the risk of antibiotic prescription (RR 2.1, 95% CI 1.44 to 3.06). Treatment decisions were considered adequate in 64.1% of the suspected UTI episodes. Overtreatment occurred more often than undertreatment. Of the inadequate treatment decisions, 29.3% was due to treatment of UTI episodes in which solely non-specific symptoms were present. A high proportion of nitrofurantoin prescriptions were incorrect in UTIs with signs of tissue invasion (54.8%), indwelling catheter-associated UTIs (37.5%), and UTIs in men (29.2%). Although this is considered inadequate, non-specific symptoms were associated with antibiotic prescription for suspected UTIs in Dutch nursing home residents and nitrofurantoin was inadequately prescribed in particular groups, such as men.

## 1. Introduction

Urinary tract infections (UTIs) are one of the most common infections in nursing home residents worldwide [1,2,3]. Consequently, (suspected) UTIs are responsible for a considerable part of the total number of antibiotic prescriptions in long-term care settings (32% to 66%) [4]. Of these prescriptions, a substantial number (up to 50%) is considered unnecessary, depending on the definition used [5,6]. 

Unnecessary antibiotic prescriptions increase the risk of adverse events at the individual level (such as *Clostridium difficile* infections and allergies [7,8]) and at the global level through antimicrobial resistance (AMR) development. AMR is highly prevalent in long-term care settings such as nursing homes, including resistance against commonly used antimicrobials to treat UTIs [4,5,9].

The prevalence of asymptomatic bacteriuria (ASB), defined as significant bacteriuria without (specific) symptomatology of a UTI, in the older population is high (up to 50%). Treatment of ASB in older adults is not considered useful since the patient is asymptomatic, and more importantly, treatment does not improve survival and it is associated with adverse events and AMR [10]. ASB is seen as an important contributor to unnecessary antibiotic prescriptions [11].

Distinguishing ASB from UTIs in nursing home residents is difficult. First, cognitive impairments limit the ability to express symptoms, including specific UTI symptoms [12]. Second, chronic genitourinary symptoms including frequency, urgency, and urinary incontinence are highly prevalent due to other underlying conditions, such as prostatic hypertrophy or neurogenic bladder [11,13]. Third, UTIs may present with non-specific symptoms such as an altered mental status [11]. Although it is commonly recognized that the presence of non-specific symptoms alone should not lead to antibiotic prescription per se, it remains an important reason for antibiotic prescriptions for suspected UTIs in nursing home residents [14,15,16]. Finally, an appropriate diagnostic test to distinguish UTIs from ASB or pyuria (presence of leukocytes without infection) is lacking, which further contributes to diagnostic uncertainty.

The recently updated Dutch guideline for UTIs in frail older adults aims to reduce inappropriate antibiotic treatment of UTIs to prevent unnecessary adverse events and to combat the occurrence of AMR [15,17]. Similar to other international guidelines [10,18], it recommends withholding antibiotic treatment and carefully monitoring the patient (“wait-and-see”) in case of ASB or non-specific symptoms. According to the guideline, antibiotic prescription is justified when a combination of specific UTI or systemic symptoms are present. Dipstick results are solely recommended to rule out UTIs when both leukocytes and nitrite are negative.

Despite the recommended guideline, inappropriate antibiotic prescriptions persists. A strategy to enhance appropriate antibiotic prescriptions in this setting is to improve the diagnostic process of suspected UTI in older adults, for example, by implementing a decision tool that automatically generates treatment advice into the electronic health record [19]. For any antibiotic stewardship strategy to improve prescribing behaviours for UTIs, more insight into antibiotic treatment decisions and influencing factors is essential. Previous studies showed that both positive dipstick results and dysuria were major resident-related factors influencing antibiotic prescribing for UTIs [20,21]. To obtain a greater understanding of these risk factors, the aim of this study was to identify resident-related factors that influence antibiotic treatment decisions in suspected as well as guideline-based UTIs and to provide an overview of the appropriateness of antibiotic treatment decisions in Dutch nursing homes.

## 2. Materials and Methods

### 2.1. Study Design and Population

This study is a secondary analysis of the PROGRESS study with the primary aim of assessing the sensitivity of blood C-reactive protein and procalcitonin, measured by point-of-care tests, to diagnose UTIs in nursing home residents. The full study protocol was published previously [22]. The data collected in this study (November 2017 to August 2019) was derived from 298 suspected UTI episodes in nursing home residents. Residents were prospectively enrolled when a UTI was suspected based on the clinical reasoning of the attending physician or nurse. Exclusion criteria were (1) suspected respiratory tract infection, (2) suspected other infection requiring antibiotics, and (3) previous enrolment in the past 30 days.

### 2.2. Outcome Measures

The main outcome of this study was the antibiotic prescribing rate per resident-related factor and the corresponding relative risks (RR). The resident-related factors were gender, the presence of an indwelling catheter, signs and symptoms (specific and non-specific), and dipstick urinalysis results (leukocytes and nitrite). Other outcome measures were (1) the proportion of adequate antibiotic treatment decisions (initiating or withholding) based on the latest Dutch UTI guideline [17], where antibiotic treatment is recommended in case of specific UTI symptoms and positive dipstick results (leukocytes or nitrite), based on a Delphi expert consensus procedure [15] and (2) the proportion of adequate antibiotic agents prescribed based on the type of UTI in accordance with the guideline (cystitis, UTI with signs of tissue invasion, or indwelling catheter-associated UTI) and susceptibility results. For cystitis, nitrofurantoin or fosfomycin is recommended. To treat men with general UTIs, catheter-associated UTIs, and UTIs with tissue invasion, the use of amoxicillin-clavulanic acid, trimethoprim/sulfamethoxazole, or ciprofloxacin is recommended. Cystitis was defined as a UTI in a female without signs of tissue invasion (fever, chills, and/or delirium) [17]. Participants, treating physicians, and nurses were not informed of urine dipstick or bacterial culture results with susceptibility testing.

### 2.3. In- and Exclusion Criteria

For the analysis of the proportion of adequate antibiotic agents based on susceptibility results, episodes were excluded if the urine was collected >24 h after initiation of antibiotic therapy to reduce the risk of false-negative urine cultures.

### 2.4. Handling Missing Data

A complete case analysis was performed. When clinical symptoms or the presence of an indwelling catheter were reported to be unknown by the attending physician or nurse, this was considered an absence of the symptom or catheter.

### 2.5. Data Analysis

For the main outcome, RR and 95% confidence intervals were calculated in order to obtain insight into the relationship between antibiotic prescription and the resident-related factors. The percentages of adequate and inadequate treatment decisions were calculated from all suspected UTI episodes in which antibiotics were prescribed. The percentages of adequate and inadequate withholding of antibiotics were calculated from all suspected UTI episodes in which no antibiotics were prescribed. The calculation of the proportion of adequate antibiotic agents prescribed based on the type of UTI in accordance with the guideline (cystitis, UTI with signs of tissue invasion, or indwelling catheter-associated UTI) was performed differently. Firstly, suspected UTI episodes were only included if they fulfilled the guideline definition of either cystitis, UTI with signs of tissue invasion or indwelling catheter-associated UTI [17]. Then, adequate antibiotic prescription, “overtreatment” (antibiotic prescription other than nitrofurantoin or fosfomycin for cystitis), and “undertreatment” (nitrofurantoin or fosfomycin for UTIs in patients with signs of tissue invasion or for indwelling catheter-associated UTIs) were calculated. The heat map was created using Python, Version 3.7.1. Python libraries Pandas, Version 1.0.3 and Numpy, Version 1.18.1 were used for data processing. Python libraries Matplotlib, Version 3.1.3 and Seaborn, Version 0.10.1 were used for visualization. All other analyses were performed using IBM SPSS, Version 26.

## 3. Results

### 3.1. Baseline Characteristics

#### 3.1.1. Nursing Home Residents

The dataset consisted of 298 suspected UTI episodes identified in 206 residents (30.9% recurrent infections) (Figure 1). The majority of the residents were female (79.1%, *n* = 163), the median age was 87 years (range 66–107), and over 70% (*n* = 146) were residents of psychogeriatric wards.

#### 3.1.2. Suspected UTI Episodes

Non-specific symptoms, with or without specific UTI symptoms, were present in the majority of episodes (72.1%, *n* = 215) (Figure 1). In 60 suspected UTI episodes (20.1%), only non-specific symptoms were present. Urine dipstick analysis results for leukocyte esterase and nitrite were positive in 226 (75.8%) and 113 (37.9%) suspected UTI episodes, respectively. In 10 UTI episodes, an indwelling catheter was present (3.4%).

#### 3.1.3. Uropathogens

In 120 of the 298 suspected UTI episodes (40.3%), an uropathogen was identified (data not shown). Among these episodes, *Escherichia coli* (*E. coli*) was most commonly often identified (*n* = 90). Other uropathogens found were *Proteus mirabilis* (*n* = 15), *Enterococcus faecalis* (*n* = 3), *Klebsiella pneumoniae* (*n* = 3), *Kelbsiella variicola* (*n* = 3), and *Klebsiella oxytoca* (*n* = 1).

### 3.2. Characteristics of Treated versus Untreated Suspected UTI Episodes

In total, 185 suspected UTI episodes were treated with antibiotics (62.1%) (Figure 1). In 8 of 10 indwelling catheter-associated UTI episodes, antibiotics were prescribed (80.0%) (Table 1).

#### 3.2.1. Signs and Symptoms

The proportion of treated UTI episodes was higher when multiple specific UTI symptoms were present: in 61.5% (40/65) of the suspected UTI episodes antibiotics were prescribed when one specific symptom was present, while in 14 out of 15 (93.3%) of the suspected UTI episodes, antibiotics were prescribed when 4 specific UTI symptoms were present (data not shown). Antibiotics were most frequently prescribed in episodes with reported dysuria (87.8%, 65/74), followed by suprapubic pain (78.4%, 69/88) (Table 1). Dysuria was associated with a 1.8-fold increase in the risk of antibiotic prescription (95% CI 1.50 to 2.10). When only non-specific symptoms were present, antibiotics were prescribed in 45.0% of the episodes (27/60).

#### 3.2.2. Urine Dipstick Results

The presence of positive leukocyte esterase results was associated with more than a two-fold increase in the risk of antibiotic prescription (RR 2.1, 95% CI 1.44–3.06) (Table 1). When both nitrite and leukocyte esterase results were positive, 81 of 100 suspected UTI episodes were treated with antibiotics (81.0%) (RR 1.5, 95% CI 1.31–1.83). When both results were negative, 12 of 44 episodes were treated with antibiotics (27.3%, RR 0.4, 95% CI 0.24–0.65).

### 3.3. Prescription Proportions in Resident-Related Factor Combinations 

In episodes presenting with a combination of dysuria and several other resident-related factors, the proportion of antibiotic prescription was high (up to 100%) (Figure 2). Especially in combination with specific UTI symptoms, leukocyte esterase results showed a pattern of increasing antibiotic prescription proportions when higher numbers of leukocytes were found. Antibiotics were prescribed in almost all suspected catheter-associated UTI episodes (*n* = 10), regardless of the additional resident-related factor.

### 3.4. Adequate Treatment Decisions

Overall, 64.1% of the treatment decisions were adequate (191/298): 50.3% of the decisions to prescribe antibiotics were adequate (93/185) and 86.7% of the decisions to withhold antibiotics were adequate (98/113) (Figure 1). Inadequate prescription occurred mostly in suspected indwelling catheter-associated UTI episodes: 87.5% (7/8) compared to 48.0% (85/177) in residents without indwelling catheters (data not shown). Of the inadequate prescriptions, 29.3% (27/92) presented with solely non-specific symptoms and lacked specific UTI symptoms. In 72.8% (67/92) and 50.0% (46/92) of the inadequate prescriptions, dipstick results were positive for leukocyte esterase and nitrite, respectively.

#### 3.4.1. Adequate Antibiotic Agents

Among the suspected UTI episodes, nitrofurantoin was the most commonly prescribed antibiotic agent (63.2%, 117/185), followed by amoxicillin-clavulanic acid (17.3%, 32/185) (data not shown). Nitrofurantoin was inadequately prescribed for seven men with suspected UTIs (29.2%). Three suspected catheter-associated UTI episodes (37.5%) and seventeen of the suspected UTI episodes (54.8%) with signs of tissue invasion were inadequately treated with nitrofurantoin.

Of the 298 suspected UTI episodes, 108 (36.2%) fulfilled the guideline’s UTI definitions: 80 cystitis (74.1%), 22 UTI with signs of tissue invasion (25.0%), and 1 indwelling catheter-associated UTI (0.9%) (data not shown). Of these guideline-based UTIs, 86.1% (93/108) were treated with antibiotics. The adequate antibiotic agent (nitrofurantoin or fosfomycin) was used in 57 of the 69 treated cystitis episodes (82.6%). In 9 of the 23 treated guideline-based UTIs with signs of tissue invasion (39.1%), the adequate antibiotic agent (amoxicillin-clavulanic acid, trimethoprim/sulfamethoxazole, or ciprofloxacin) was prescribed.

#### 3.4.2. Adequate Antibiotic Agents Based on Susceptibility Results

Among all treated suspected UTI episodes with urine specimens collected <24 h after initiating antibiotics and a positive urine culture, 84.5% (87/103) of the identified uropathogens were susceptible to the prescribed antibiotic agent (data not shown). *E. coli* isolates were mostly susceptible (92.3%, 72/78), while *P. mirabilis* isolates were only susceptible in 6 of 13 (46.2%) UTI episodes for the antibiotic agents prescribed.

## 4. Discussion

This study shows that dysuria and positive leukocyte esterase dipstick results increased the risk of antibiotic prescription for suspected UTI in Dutch nursing homes. Moreover, the proportion of treated UTI episodes increased when increasing numbers of leukocytes were found, especially in combination with the presence of specific UTI symptoms. These data confirm that, against guideline recommendations, the presence of non-specific symptoms resulted in treatment of a UTI. A third of the antibiotic treatment decisions were either inadequately prescribed or inadequately withheld, according to the Dutch guideline for UTIs in frail older adults [17]. Overtreatment (49.7%) seems to be a bigger problem than undertreatment (13.3%). Notably, undertreatment due to prescription of an inadequate antibiotic agent was prominent in suspected as well as guideline-based UTIs with signs of tissue invasion, indwelling catheter-associated UTIs and UTIs in men.

The reluctance to prescribe antibiotics for only non-specific symptoms is not new. Previous guidelines, including the former Dutch guideline, recommend refraining from antibiotic treatment for episodes without specific UTI symptoms [10,18,23]. Therefore, it is surprising that solely non-specific symptoms were still present in 20% of the suspected UTI episodes and almost half of them were treated with antibiotics. In comparable studies in the US, 40–50% of the antibiotic prescriptions occurred in suspected UTIs with solely non-specific symptoms [24,25]. Thus, guideline recommendations alone do not prevent treatment of non-specific symptoms and additional approaches to improve antibiotic prescribing behaviour are warranted. Involvement, education, and coaching of both physician and nursing staff have already been presented as strategies to improve antibiotic stewardship [19,26,27,28,29,30,31]. Additional strategies include providing communication tools to nurses to improve effective communication among healthcare workers [32] and interventions directed at appropriate urine culturing [33]. Supplying feedback to physicians and nurses by using the tools provided by this study could actively enhance awareness of their own prescription behaviour and thereby improve guideline adherence in daily practice.

We found that positive leukocyte esterase results were associated with a more than doubled risk of antibiotic prescription. This is in line with an earlier study in nursing homes that reported a positive dipstick result as the most important resident characteristic prognosticating antibiotic prescription for suspected UTIs [20]. On the other hand, in our study, antibiotics were prescribed in nearly a third of the suspected UTI episodes with negative nitrite and leukocyte esterase dipstick results despite the high negative predictive value of dipstick analysis [17,34]. Despite efforts to find another suitable test to diagnose UTIs in nursing home residents [35,36], dipstick testing remains the most important diagnostic tool for UTIs in nursing homes. The diagnostic role of dipstick testing—solely to rule out UTIs—may therefore be reinforced in future guidelines.

In our study, inadequate treatment decisions were made in approximately one third of the suspected UTIs and overtreatment occurred more often than undertreatment, which is in concordance with previous findings in Dutch nursing homes [5]. Surprisingly, by assessing the appropriateness of antibiotic prescription for different types of UTIs, our study revealed high rates of incorrect small-spectrum (nitrofurantoin) use in UTIs with signs of tissue invasion, indwelling catheter-associated UTIs, and UTIs in men. This undertreatment in terms of antibiotic agent is in contrast with previous studies, where up to 90% incorrect use of broad-spectrum antibiotics in lower UTIs was described [6,37]. Contrarily, the proportion of adequate prescriptions in cystitis (nitrofurantoin or fosfomycin) in our study was remarkably higher. Possible explanations for these differences are (1) a lack of recommendations regarding the antibiotic agent in upper, lower, and catheter-associated UTIs and men in the previous Dutch guideline for UTIs in frail older adults and (2) the recommendation of nitrofurantoin as treatment for lower UTI in men and catheter-associated UTIs in the Dutch College of General Practitioners guideline [23,38]. Treatment of upper UTIs with antibiotics such as nitrofurantoin induce therapy failure since these agents reach high urine but low serum levels [39]. Although the recent guideline generally concentrates on reducing overtreatment, an additional focus on undertreatment, especially in terms of adequate antibiotic agents, could improve antibiotic treatment decisions in this frail population.

In our study, *E. coli* comprised the majority of the isolated uropathogens and were susceptible to the prescribed therapy in 92.3% of UTI episodes. The overall uropathogen susceptibility in this study was higher (84.5%) than the uropathogen susceptibility in an American nursing home study (77.4%) [40].

A strength of this study is that it provides an overview of the individual and combined influence of a wide range of resident-related clinical and demographic factors in antibiotic treatment decisions in suspected as well as guideline-based UTIs. In addition, because of our prospective study design in which residents were included based on the clinical suspicion of a UTI, both treatment decisions to prescribe and to withhold antibiotics were assessed. Hereby, this study gives a complete overview of the management of suspected UTIs in nursing homes in terms of overtreatment and undertreatment. Previous studies in this field had a retrospective study design in which the treatment decision to prescribe antibiotics could only be assessed [1,24] or in which they were unable to assess the influence of each individual factor on the occurrence of antibiotic prescription due to unspecified clinical situations or qualitative approaches [21]. Its multicentre approach, including different nursing home organizations and different types of wards in which differences in prescribing behaviour and culture may exist further enhances the generalizability of this study.

The main limitation of this study is that more than two thirds of the UTI episodes were enrolled before publication of the new Dutch guideline for UTIs in frail older adults and the remaining episodes were enrolled within a year after publication. Although the proportions of adequate treatment decisions did not differ between the subgroups enrolled before and after guideline publication, this may have resulted in an overly stringent assessment of the proportion of adequate treatment decisions. However, the previous guideline already recommended careful exclusion of other infections in case of non-specific symptomatology before initiation of antibiotics [23]. Second, it solely assessed the influence of resident-related demographic and clinical factors in antibiotic treatment decisions. Prior investigations of these decisions in older adults in long-term care facilities have noted the importance of the complex interplay between a variety of rational and non-rational influencing factors such as advance care plans; availability and utilization of diagnostic resources; and the influence of colleagues, nurses, or the resident’s family [21]. These qualitative data elucidated the possible underlying factors of overtreatment, which may suggest additional intervention targets improve appropriate prescribing in this setting. Furthermore, since inclusion in this study required proactive registration by physicians or nurses, untreated patients may have been less likely to be included, contributing to altered proportions of (in)adequate antibiotic withholding.

## 5. Conclusions

This study showed that unnecessary antibiotic prescriptions in suspected UTI episodes were driven by dipstick results and the presence of solely non-specific symptoms, and it revealed inadequate nitrofurantoin use in particular risk groups, such as men. When re-evaluating the current guidelines, a greater focus on these pitfalls causing overtreatment as well as undertreatment should therefore be considered. Due to the interdependence of symptoms and symptom complexes characterising UTIs, causality between each resident-related factor and the occurrence of antibiotic prescription could not be determined. A principal component analysis with a larger sample could further determine the causal relationship between antibiotic prescription and its influencing factors. The results of this study could, however, encourage a greater focus on guideline adherence. Furthermore, it could be used for educational matters targeted at nursing home nurses and physicians to reduce the gap between guideline recommendations and daily practice to improve antibiotic treatment decisions in this frail population.

## Figures and Tables

**Figure 1 antibiotics-11-00140-f001:**
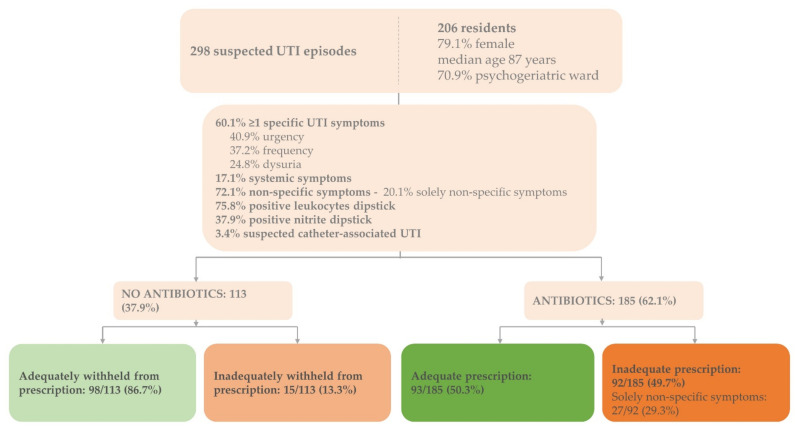
Baseline characteristics and adequate antibiotic prescriptions for suspected UTIs in Dutch nursing home residents. Baseline characteristics of the enrolled residents (*n* = 206) and suspected UTI episodes (*n* = 298) including proportions of adequate and inadequate treatment decisions according to Dutch guideline for UTIs in people who are frail and older [17]; adequate prescription: adequate prescription of antibiotics based on signs and symptoms justifying antibiotics; inadequate prescription: inadequate prescription based on signs and symptoms justifying antibiotics; adequately withheld: antibiotics adequately withheld based on signs and symptoms justifying antibiotics; inadequately withheld: antibiotics inadequately withheld based on signs and symptoms justifying antibiotics; UTI = urinary tract infection.

**Figure 2 antibiotics-11-00140-f002:**
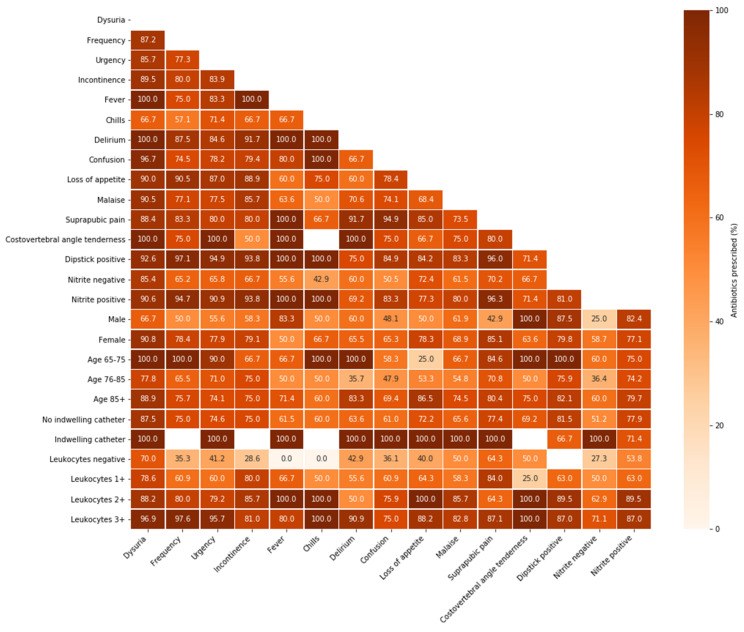
Antibiotic prescription in resident-related factor combinations for suspected UTI episodes in Dutch nursing home residents. Proportions of antibiotic prescriptions in the enrolled suspected UTI episodes (*n* = 298) per resident-related factor combination (both on x- and y-axis); light colours represent lower prescription proportions, while dark colours represent higher prescription proportions; UTI = urinary tract infection.

**Table 1 antibiotics-11-00140-t001:** Antibiotic prescription for suspected UTI episodes in Dutch nursing home residents.

	Resident-Related Factor	Antibiotic Prescription (*n* = 185)	RR [95% CI]
Gender	Female vs. Male	65.4% (*n* = 161)/46.2% (*n* = 24)	1.4 [1.04–1.93]
Indwelling catheter	Yes vs. No	80.0% (*n* = 8)/61.3% (*n* = 174)	1.3 [0.95–1.80]
Specific UTI symptoms	Dysuria Yes vs. No Urgency Yes vs. No Frequency Yes vs. No Urinary incontinence Yes vs. No Urethral discharge Yes vs. No	87.8% (*n* = 65)/49.5% (*n* = 93) 74.6% (*n* = 91)/49.3% (*n* = 71) 74.8% (*n* = 83)/50.0% (*n* = 78) 74.5% (*n* = 41)/57.7% (*n* = 127) 42.9% (*n* = 3)/61.9% (*n* = 159)	1.8 [1.50–2.10] 1.5 [1.24–1.84] 1.5 [1.24–1.81] 1.3 [1.07–1.56] 0.7 [0.29–1.64]
Systemic symptoms	Fever Yes vs. No Chills Yes vs. No Delirium Yes vs. No	64.3% (*n* = 9)/60.7% (*n* = 164) 60.0% (*n* = 6)/61.7% (*n* = 174) 64.7% (*n* = 22)/61.8% (*n* = 144)	1.1 [0.71–1.58] 1.0 [0.58–1.63] 1.0 [0.80–1.37]
Additional symptoms	Costovertebral angle tenderness Yes vs. No Suprapubic pain Yes vs. No	69.2% (*n* = 9)/61.7% (*n* = 153) 78.4% (*n* = 69)/51.1% (*n* = 94)	1.1 [0.77–1.63] 1.5 [1.28–1.84]
Non-specific symptoms	Confusion vs. No confusion Loss of appetite Yes vs. No Malaise Yes vs. No	62.6% (*n* = 107)/62.4% (*n* = 73) 73.2% (*n* = 41)/57.5% (*n* = 130) 67.4% (*n* = 64) 58.2% (*n* = 110)	1.0 [0.84–1.20] 1.3 [1.05–1.55] 1.2 [0.96–1.39]
Solely non-specific symptoms		45.0% (*n* = 27)	
Leukocytes	Positive * vs. Negative	69.9% (*n* = 158)/33.3% (*n* = 19)	2.1 [1.44–3.06]
	Positive * leukocytes - 1+ - 2+ - 3+	54.8% (*n* = 40) 72.2% (*n* = 39) 79.8% (*n* = 79)	
Nitrite	Positive vs. Negative	77.9% (*n* = 88)/52.4% (*n* = 89)	1.5 [1.25–1.77]
Nitrite & leukocytes	Positive	81.0% (*n* = 81)	1.5 [1.31–1.83]
Nitrite & leukocytes	Negative	27.3% (*n* = 12)	0.4 [0.24–0.65]

The proportion of antibiotic prescriptions in the enrolled suspected UTI episodes (*n* = 298) per resident-related factor and RR evaluating the association between antibiotic prescription and the resident-related factors; * leukocytes positive: 1+ = 10–25 leukocytes per μL, 2+ = ~75 leukocytes per μL, 3+ = ~500 leukocytes per μL); missing data (n): indwelling catheter (4), dysuria (36), urgency (32), frequency (31), urinary incontinence (23), urethral discharge (34), fever (14), chills (6), delirium (31), costovertebral angle tenderness (37), suprapubic pain (26), confusion (10), loss of appetite (16), malaise (14), dipstick leukocytes and nitrite (15); CI = confidence interval; RR = relative risk; UTI = urinary tract infection.

## Data Availability

The data in this study are available from the corresponding author upon request.

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
