# Peer review of "Resident-Related Factors Influencing Antibiotic Treatment Decisions for Urinary Tract Infections in Dutch Nursing Homes"

_antibiotics, 2022, doi:10.3390/antibiotics11020140_

Round 1
Reviewer 1 Report
The manuscript is interesting and well structured. It can be useful for the treatment of UTI. The English is quite fluent. However, two minor corrections are needed:
- On line 83 the authors write “Other outcome measures were (1) the proportion adequate antibiotic treatment decisions (initiation or withholding) based on the latest Dutch UTI guideline [15], where antibiotic treatment is recommended in case of specific UTI symptoms and positive dipstick results (leukocytes or nitrite), based on a Delphi expert consensus procedure [14] and (2) the proportion adequate antibiotic agents prescribed based on the type of UTI in accordance with the guideline (cystitis, UTI with signs of tissue invasion or indwelling catheter-associated UTI), and susceptibility results”. Please the authors modify the sentence as follows: “Other outcome measures were (1) the proportion of adequate antibiotic treatment decisions (initiation or withholding) based on the latest Dutch UTI guideline [15], where antibiotic treatment is recommended in case of specific UTI symptoms and positive dipstick results (leukocytes or nitrite), based on a Delphi expert consensus procedure [14] and (2) the proportion of adequate antibiotic agents prescribed based on the type of UTI in accordance with the guideline (cystitis, UTI with signs of tissue invasion or indwelling catheter-associated UTI) and susceptibility results”;
- On line 290 the authors write “Prior investigations of these decisions in older adults in long-term care facilities have noted the importance of the complex interplay between a variety of rational and non-rational influencing factors such as advance care plans, availability and utilization of diagnostic resources and the influence colleagues, nurses or the resident’s family [18]”. Please the authors modify the sentence as follows: “Prior investigations of these decisions in older adults in long-term care facilities have noted the importance of the complex interplay between a variety of rational and non-rational influencing factors such as advance care plans, availability and utilization of diagnostic resources and the influence of colleagues, nurses or the resident’s family [18]”.
Author Response
Dear editor, dear reviewers,
We thank you for your review responses and for considering our manuscript entitled “Resident-related factors influencing antibiotic treatment decisions for urinary tract infections in Dutch nursing homes” for publication in Antibiotics.
Please find attached a second revised version of our manuscript including the revised parts (highlighted). The list of the point to point responses to your questions and comments per reviewer can also be found below, please see the attachment.
We thank the editorial team for their second review of our manuscript and the additional useful comments and questions, which hopefully have improved our manuscript considerably.
Sincerely,
On behalf of all co-authors,
Lisa Marie Kolodziej

Reviewer 2 Report
The research presents data that confirms against guideline recommendations the presence of non-specific symptoms resulted in treatment of an UTI, with high level of overtreatment and low level of undertreatment.
The second paragraph from Introduction could reference for Clostridium difficile the article
https://www.ncbi.nlm.nih.gov/pmc/articles/PMC8233718/
Please replace Figure 1 with a sharper one.
Can you specify if antibiograms were used in antibiotic prescription?
Author Response

(The authors gave the same response as above.)

Reviewer 3 Report
This paper study the resident-related factors influencing antibiotic treatment decisions for urinary tract infections in Dutch nursing homes. The aim of this cohort study was to identify resident-related factors that influence antibiotic treatment decisions for urinary tract infections (UTIs) in nursing home residents and to provide an overview of the appropriateness of antibiotic treatment decisions according to the updated Dutch guideline for UTIs in frail older adults. There are a few weaknesses that should be addressed in this paper. Therefore, I suggest the authors resubmit it after a major revision. My suggestions are as follows:
- Authors should enrich the literature review by addressing more relevant papers.
- In lines 55-56-57 you mentioned: "Like other international guidelines[9, 16], it recommends to withhold antibiotic treatment and carefully wait-and-see in case of ASB or non-
specific symptoms" please explain more about it....It is unclear. - Comparisons with existing approaches are missing.
- The data analysis subsection is too short. Please explain more or provide a statistical analysis of your suggested data.
- sub-sections 3.1.1, 3.1.2 and 3.1.3 are not visible. Please edit this part and explain more.
- Instead of subsection 3.1.3 in line 122, you consider 3.1.1 again mistakenly. Please edit this part.
- In sub-section 3.3 in line 168, you mentioned: "In episodes presenting with a combination of dysuria and several other resident-related factors, the proportion of antibiotic prescription was high (up to 100%) (Figure 2). Especially in combination with specific UTI symptoms, leukocyte esterase results showed a pattern of increasing antibiotic prescription proportions when higher numbers of leukocytes were found. Regardless of the additional resident-related factor, antibiotics were prescribed in almost all suspected catheter-associated UTI episodes" Please explain more. What is your mean by mentioning the catheter factor?
- How can you describe the advantage of your approach to the problem? You should compare this approach with others in the literature. Please add at least 10 relevant recent references(2020-2021).
Author Response

(The authors gave the same response as above.)

Round 2
Reviewer 3 Report
The authors have replied to all my concerns and have considered all my comments. This version is acceptable and available for publication.